# Restoration of normal central pain processing following manual therapy in nonspecific chronic neck pain

Josu Zabala Mata 1,2☯*, Jon Jatsu Azkue2☯, Joel E. Bialosky3,4‡, Marc Saez5,6‡, Estíbaliz Dominguez López2‡, Maialen Araolaza Arrieta 1‡, Ion Lascurain-Aguirrebeña7☯

1 Department of Physical Therapy, Deusto Physical Therapiker, Faculty of Health Science, University of Deusto, San Sebastian, Spain, 2 Department of Neurosciences, Faculty of Medicine and Nursing, University of the Basque Country UPV/EHU, Leioa, Spain, 3 Department of Physical Therapy, University of Florida, Gainsville, Florida, United States of America, 4 Clinical Research Center, Brooks Rehabilitation, Jacksonville, Florida, United States of America, 5 Research Group on Statistics, Econometrics and Health (GRECS), University of Girona, Girona, Spain, 6 CIBER of Epidemiology and Public Health (CIBERESP), Madrid, Spain, 7 Physiotherapy, Department of Physiology, Faculty of Medicine and Nursing, University of the Basque Country UPV/EHU, Leioa, Spain

☯ These authors contributed equally to this work.
‡ JEB, MS, EDL and MAA also contributed equally to this work.
* josu.zabala@deusto.es

**Data Availability Statement:** All relevant data are within the manuscript and its Supporting Information files.

## Abstract

### Objective

To determine if a 4-week manual therapy treatment restores normal functioning of central pain processing mechanisms in non-specific chronic neck pain (NSCNP), as well as the existence of a possible relationship between changes in pain processing mechanisms and clinical outcome.

### Design

Cohort study.

### Methods

Sixty-three patients with NSCNP, comprising 79% female, with a mean age of 45.8 years (standard deviation: 14.3), received four treatment sessions (once a week) of manual therapy including articular passive mobilizations, soft tissue mobilization and trigger point treatment. Pressure pain thresholds (PPTs), conditioned pain modulation (CPM) and temporal summation of pain (TSP) were evaluated at baseline and after treatment completion. Therapy outcome was measured using the Global Rating of Change Scale (GROC), the Neck disability Index (NDI), intensity of pain during the last 24 hours, Tampa Scale of Kinesiophobia (TSK) and Pain Catastrophizing Scale (PCS). Two sets of generalized linear mixed models with Gaussian response and the identity link were employed to evaluate the effect of the intervention on clinical, psychological and psychophysical measures and the association between psychophysical and clinical outcomes.

**Funding:** The author(s) received no specific funding for this work.

**Competing interests:** The authors have declared that no competing interests exist.

## Results

Following treatment, an increased CPM response (Coefficient: 0.89; 95% credibility interval = 0.14 to 1.65; P = .99) and attenuated TSP (Coefficient: -0.63; 95% credibility interval = -0.82 to -0.43; P = 1.00) were found, along with amelioration of pain and improved clinical status. PPTs at trapezius muscle on the side of neck pain were increased after therapy (Coefficient: 0.22; 95% credibility interval = 0.03 to 0.42; P = .98), but not those on the contralateral trapezius and tibialis anterior muscles. Only minor associations were found between normalization of TSP/CPM and measures of clinical outcome.

## Conclusion

Clinical improvement after manual therapy is accompanied by restoration of CPM and TSP responses to normal levels in NSCNP patients. The existence of only minor associations between changes in central pain processing and clinical outcome suggests multiple mechanisms of action of manual therapy in NSCNP.

## Introduction

Neck pain is among the top five causes of disability in middle- and high-income countries and among the top ten as a cause of global disability [1]. Despite investment in research, the prevalence of neck pain has not declined substantially in the last two decades [2]. Since little relationship with radiological findings and no specific cause is found to explain symptoms, patients are usually classified as suffering from non-specific neck pain (NSCNP) [3].

Guidelines advocate treating patients with NSCNP with exercise and manual therapy [4, 5]. However, systematic reviews assessing clinical outcomes of manual therapy, including for example cervical manipulation, thoracic manipulation, cervical mobilization and massage, report low to moderate treatment effects at best [6]. Such relatively modest benefit from current therapeutic interventions should be of no surprise, in light that critical aspects of treatment remain to be established, such as optimal dosage and clinical parameters, best indicated forms of mobilization, and possible target patient subpopulations. This may be partly due to the fact that mechanisms of action of manual therapy are not yet fully understood. Although biomechanical effects, neural hysteresis, and segmental neurological modulation have long been postulated as underlying mechanisms of action of manual therapy, hypotheses have in recent years shifted towards a potential role of altered pain processing in the central nervous system [7].

Inter-individual variability in the functioning of central pain processing mechanisms has been postulated as an alternative framework to understand heterogeneity of treatment outcomes [8]. Several studies have reported disturbances in central pain processing in patients with NSCNP [9–11]. A meta-analysis has confirmed the occurrence of hyperalgesia distal to the most painful site [12], a probable indication of the occurrence of central sensitization in the NSCNP population. Central sensitization (CS) refers to a state of increased central responsiveness to nociceptive inputs associated with plastic changes in nociceptive circuits and pathways [13]. There is consistent evidence of altered central pain processing in patients with NSCNP, including both pronociceptive and antinociceptive mechanisms. Temporal summation of pain (TSP), a gradual increment of the pain sensation elicited by repeated C-fiber–mediated stimuli which is evaluated as a measure of pronociceptive mechanisms, is enhanced

in NSCNP patients [9, 11, 14]. In addition, disruption of endogenous antinociception has also been found, such as the impairment of the so-termed Conditioned Pain Modulation (CPM) [10, 11].

Although changes in central pain processing have been reported following manual therapy (Mulligan's mobilization with movement, cervical manipulation, anteroposterior mobilisations, lateral glide mobilization, Maitlands passive accesory mobilization, etc.), most studies have relied on static psychophysical measures (largely assessment of pressure pain thresholds) and found amelioration of local [12, 15, 16] and in some cases distal hyperalgesia [17]. However, studies assessing the effects of manual therapy using dynamic psychophysical tests are scarce, and relatively little is known on the effects of manual therapy on central pain processing mechanisms. Dynamic psychophysical tests have been postulated to better assess pain processing in the central nervous system [18] as they evaluate central processing systems rather than pain perception. Although a systematic review showed that physical therapy may reverse alterations in pain processing that accompany several musculoskeletal conditions [19], few studies have specifically addressed the effect of manual therapy, and, of these, only one included patients with NSCNP [20]. This latter study evaluated the effect of neurodynamic upper limb mobilizations, and found beneficial effects of therapy on CPM but not on TSP. Other studies that have assessed the effect of physiotherapy other than manual therapy in NSCNP have found no effects of therapeutic exercise and virtual reality on TSP [21, 22] and CPM [22]. Currently, the effect of manual therapy on central pro- and antinociceptive processing in patients with this condition is unknown.

For restoration of normal central pain processing to be considered as a potential mechanism of action of manual therapy, it should be associated with improvements in clinical status [7]. The only studies that have so far addressed this issue failed to find associations between clinical improvement and changes in mechanical pain thresholds [23, 24], and none has assessed a possible association between restoration of normal central pain processing and clinical outcomes of manual therapy.

The present study directly addresses the previously identified research gaps of 1) failure to determine the effects of manual therapy on dynamic measures of quantitative sensory testing i.e. TSP and CPM and 2) failure to link changes in pain sensitivity processing to clinical outcomes. Consequently, we aimed to determine whether manual therapy restores normal functioning of central pain processing mechanisms in patients with NSCNP. As a secondary aim, we sought to evaluate the relationship between clinical outcome and changes in central pain processing mechanisms following manual therapy.

## Methods

A single-center, prospective study was conducted at a primary care physiotherapy clinic in the Bizkaia region of Spain between March 2020 and July 2021. All patients provided written consent before data collection and their rights were protected. The study was approved by the institutional review board at the University of the Basque Country–UPV/EHU (Ethical approval reference: M10_2018_160MR1_ZABALA MATA) and registered before study commencement (ClinicalTrials.gov record number: ACTRN12620000163909).

### Participants

In a two-sided test, assuming an alpha of 5% and an statistical power of 80% a sample size of 63 subjects (observed longitudinally up to 3 times) were required to detect a minimum difference of 10% in TSP and CPM measures [25]. A difference of less than 10% was considered negligible. We used the 'pwr' package, in the free statistical environment R (version 4.3.2), based

on the formula provided by Cohen [25, 26]. People seeking treatment for NSCNP at a primary care physiotherapy clinic were invited to participate. Individuals were included in the study if they met the following inclusion criteria: pain of mechanical origin (i.e. pain is reproduced by neck movements or positions) and non-traumatic (insidious) onset. They were excluded if they presented: a whiplash associated disorder pathology; widespread, non-anatomical distribution of pain; stimulus-independent spontaneous pain; neurological (either sensory or motor) deficit; radicular pain; had undergone or were awaiting neck surgery, or referral to other health professional to exclude non- musculoskeletal causes of their neck pain (ex. cancer) was required.

## Clinical assessment

Age, sex, height, and weight were recorded from participants, and patients completed the Neck Disability Index (NDI), the Pain Catastrophizing Scale (PCS) and The Tampa Scale of Kinesophobia (TSK) questionnaires. The NDI is the most frequently used, self-administered questionnaire for assessing cervical disability. The questionnaire consists of 10 items on activities of daily living, and each item is scored from 0 to 5, where higher scores indicate greater disability. It has demonstrated good to excellent internal consistency and moderate to excellent test-retest reliability [27]. The validated Spanish version was used [28]. The PCS is a 13-item questionnaire that measures catastrophic thoughts and feelings about pain. Total scores range from 0 to 52, and higher scores indicate higher levels of pain-related catastrophizing. This questionnaire has demonstrated high internal consistency and discriminative validity in adult community and pain outpatient samples [29]. The validated Spanish version was used [30]. Pain-related fear of movement was assessed using the 11-item TSK; scores on each item range from 1 to 4, where higher scores are indicative of greater fear [31]. This questionnaire has exhibited satisfactory internal consistency and demonstrated significant validity [32], including its validated Spanish version, which was used for the current study [33].

Maximum and mean intensities of pain experienced over the last 24 hours, and pain experienced during neck movements (flexion, extension, right and left rotation and side flexion) were recorded using a 0–10 numeric rating scale anchored with 0 = no pain at all to 10 = worst pain imaginable. Patients were also asked to complete the Patient Specific Functional Scale (PSFS) [34], a self-reported measure of perceived level of disability on specific items relevant for them. The PSFS has excellent reliability and moderate to strong validity [35].

In addition, patients were asked to rate their perceived treatment effect using the Global Rating of Change Scale (GROC). The GROC is a 15-point scale where clinical change is rated from -7 (a very great deal worse), through 0 (no change), to +7 (a great deal better) [36]. This questionnaire has shown excellent reliability [37].

All clinical measures except GROC (recorded only post-treatment) were obtained in single sessions both at baseline and after treatment completion. A maximum of 24 hours elapsed both from the first clinical assessment session to treatment initiation, and from the last treatment session to the second clinical assessment.

## Psychophysical assessment

Pressure Pain Thresholds (PPT), defined as the minimum pressure at which pressure sensation becomes a painful sensation [38], were measured at several locations using a digital hand-held algometer with a 1-cm$^2$-diameter rubber tip (Fisher, Pain Diagnostics and Thermography Inc, Great Neck, NY, USA). For local assessment of pain sensitivity, PPTs were measured bilaterally at the angle of the upper trapezius fibers, 5 and 8 cm above and medial to the superior angle of the scapula, and remote sensitivity was assessed on the tibialis anterior at a location 2.5 cm

lateral and 5 cm inferior to the anterior tibial tuberosity. Subjects were instructed to report their first perceived pain sensation during an incremental pressure application at 1 kg/ sec. The same procedure was repeated three times, 1 min apart, and the mean of three measurements was used for analysis. Patients were familiarized with the measurement protocol prior to the actual measurements. This procedure has shown high reliability in neck pain patients [39].

For assessing TSP, patients were seated in a quiet room with their hand rested on a table (same side as neck pain, or the side of most painful neck pain in patients with bilateral pain) and two adhesive Ag/AgCl electrodes were placed on the hand dorsum, 2 cm apart. Electrical stimuli consisting of brief bursts of five, 1 ms-long positive-square pulses, were generated by a constant current electrical stimulator (DS7; Digitimer Ltd, Welwyn Garden City, UK) and delivered at 200 Hz [40], which were perceived by the participant as single stimuli. Electrical pain thresholds were first determined using the increasing and decreasing staircase method with 0.2 mA stimulus increments/decrements. The electrical pain threshold was defined as the minimum current intensity evoking a sensation rated as painful in an incremental series or the current intensity no longer evoking pain in a decremental series, and the final value was recorded as the mean of three consecutive incremental and three decremental measures. For assessing TSP, a single stimulus was administered at 1.2 times the electrical pain threshold intensity, and the participant was asked to rate the evoked pain sensation on a 0–100 numeric scale where 0 denotes no pain at all and 100 indicates the worst pain imaginable. Two minutes thereafter, 5 consecutive stimuli of the same current intensity were delivered at a frequency of 2 Hz (2.5-millisecond total stimulus duration), and the participant was asked to rate the pain sensation evoked by the stimulus perceived as the most painful. The ratio of the second rating to the first was used as the TSP measure [41]. A higher ratio was indicative of greater TSP. This protocol has been previously used [42] and is based on well-known parameters for evaluating TSP [40].

For CPM assessment, PPT was measured first on the trapezius muscle as above, and the participant was then asked to immerse his/her contralateral foot in cold water (kept at 10˚ C) for 2 minutes or until pain became unbearable. Immediately thereafter, the PPT was measured again at the same location. The CPM response was obtained by subtracting the second measure from the first [43]. A greater value was indicative of higher endogenous pain inhibition. This procedure has demonstrated good to very good reliability [44].

## Intervention

Patients received a 45-minute session of manual therapy once a week for 4 weeks. Treatments consisted of articular passive mobilizations, soft tissue mobilization, and trigger point treatment performed by the clinician following clinical reasoning. Passive mobilization treatment consisted of passive, low-speed movements performed on hypomobile and pain-reproducing spinal segments in the cervical and thoracic spine [36], including grade II–III posterior-anterior and/or antero-posterior mobilizations following the movement plane of the cervical zygapophyseal joints (upslope and downslope mobilizations) [45–48], with the patient in a supine position (Fig 1). All passive mobilizations were performed using oscillatory techniques, comprising sets of 6 oscillations. The procedure continued until the hypomobile segments regained motion, or alternatively, a maximum of 4 sets was reached. The direction and intensity of the technique were determined by the clinician based on prior clinical assessment. Soft tissue mobilization (gentle longitudinal and transverse stroking) of neck muscles was administered in order to improve connective tissue function and reduce myofascial pain [49]. This was accompanied by a trigger point ischemic compression technique on neck muscles where appropriate. In this technique, the therapist gradually applied increasing pressure to trigger

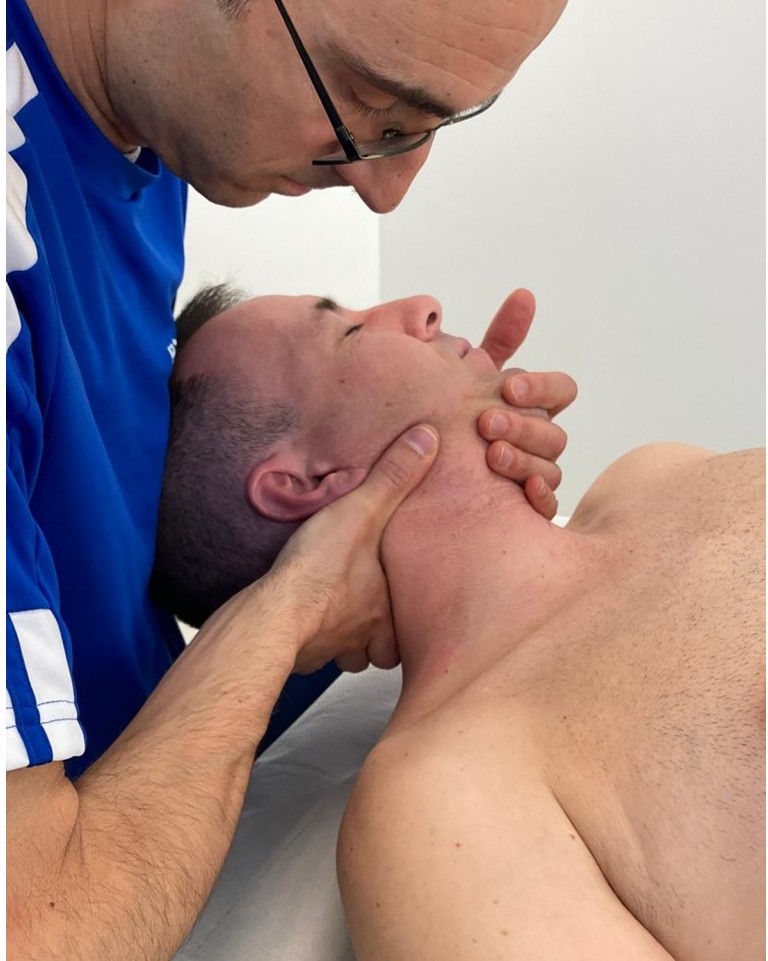

**Fig 1. Upslope and downslope mobilization.**

points until the onset of pain. Pressure was sustained until pain was relieved or the treatment surpassed one minute, whichever occurred first. Subsequently, the pressure was increased until discomfort was felt again. The therapist repeated this procedure approximately three–four times [50]. This procedure has been found effective to reduce muscular pain in NSCNP [51]. All treatments were administered by the same physiotherapist with postgraduate training and 15 years of experience in musculoskeletal physiotherapy, which was blinded to the baseline and post-treatment clinical and neurophysiological assessments.

## Statistical analysis

Two sets of generalized linear mixed model (GLMM), with Gaussian response and the identity link (i.e. equivalent to a linear regression) and heteroskedastic variance (since subjects were observed more than once), were used to assess the effect of the intervention on clinical, psychological and psychophysical measures, and the association between treatment-induced psychophysical changes and clinical and psychological outcomes. Analyses were controlled for sex, age, BMI, baseline value of the variables of interest, and individual heterogeneity. Individual heterogeneity, controlled for including a random effect, collects unobserved invariant variables over time that are specific to each individual participant, i.e. residual confounding. Such random effect also controlled the dependence between observations, since subjects were

observed at least twice (in some variables three). Given the complexity of the models, we performed inferences using a Bayesian framework. In particular, we followed the Integrated Nested Laplace Approximation (INLA) approach [52, 53]. In addition to the coefficient estimators and their 95% credibility intervals, the probability of the coefficient estimator (an absolute value being more than 1 (Prob(|estimator|)>1), Prob, was also computed (note that this is unilateral and may not coincide with the credibility interval). Unlike the p-value in a frequentist approach, this probability allows us to make inferences about associations between dependent and independent variables. For the sake of simplicity, Prob values exceeding 0.95 are equivalent to $p < .05$ in a non-Bayesian context. All analyses were conducted using the open access software *R* (version 4.2.2) [54] available through the INLA package [52, 53, 55]

## Results

Sixty-three participants took part in the study between 03/03/2020 and 21/07/2021. Demographics and baseline clinical characteristics are shown in **Table 1**. All participants attended the scheduled therapy sessions and completed the treatment, and there were no drop outs (**Fig 2**).

### Changes in pain processing, and clinical and psychological outcomes after treatment

Patients showed an improvement in central pain processing following manual therapy. Namely, the intervention both attenuated TSP response (Coefficient: -0.63; 95% credibility interval = -0.82 to -0.43; P = 1.00) and improved conditioned modulation of pain (Coefficient: 0.89; 95% credibility interval = 0.14 to 1.65; P = .99). In addition, manual therapy increased PPT on the trapezius muscle on the side of neck pain (Coefficient: 0.22; 95% credibility interval = 0.03 to 0.42; P = .98), but not on the contralateral trapezius (Coefficient: 0.01; 95% credibility interval = -0.19 to 0.21; P = .54) or the tibialis anterior muscle (ipsilateral Coefficient: -0.03; 95% credibility interval = -0.29 to 0.22; P = .59 on the side of neck pain. Contralateral coefficient: 0.01; 95% credibility interval = -0.26 to 0.29; P = .54 contralaterally) (Table 3).

Clinical pain was also ameliorated following manual therapy, as shown by the reduction in mean pain ratings at 24 hours (Coefficient: -2.52; 95% credibility interval = -2.92 to -2.13;

**Table 1. Demographics and clinical characteristics.** Values are mean (SD), number of cases or percentage as outlined below.

| N | 63 |
|---|---|
| Sex (f/m) | 50(79%)/13 |
| Age (y) | 45.8 (14.3) |
| BMI | 23.5 (3.2) |
| Neck pain duration (y) | 6.7 (5.2) |
| Mean pain 24 hours (0–10) | 4.72 (1.83) |
| Maximum pain 24 hours (0–10) | 6.26 (1.82) |
| NDI (0–50) | 11.56 (5.23) |
| PCS (0–52) | 15.38 (9.28) |
| TSK (0–44) | 23.99 (7.29) |
| PSFS (0–10) | 4.24 (1.93) |
| Pain on movement | 3.48 (2.03) |

Abbreviations: BMI, body mass index; NDI, neck disability index; PCS, pain catastrophizing scale; TSK, Tampa scale kinesophobia; PSFS, Patient specific functional scale; TSP, temporal summation pain; CPM, conditioned pain modulation.

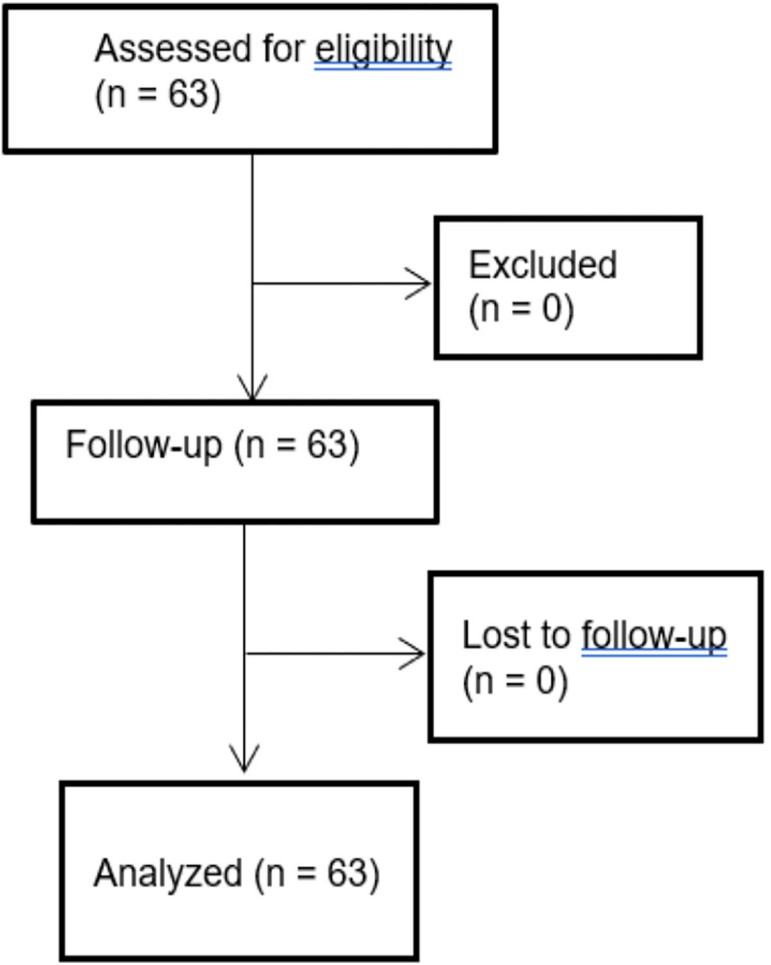

**Fig 2. Flow diagram.**

P = 1.00), maximal pain ratings at 24 hours (Coefficient: -3.07; 95% credibility interval = -3.54 to -2.59; P = 1.00) and pain ratings during neck movements (Coefficient: -2.19; 95% credibility interval = -2.51 to -1.87; P = 1.00) (Table 3). The majority of patients reported feeling "a very great deal better" or "a great deal better" (21% and 33% respectively) following treatment, 16% reported feeling "quite a bit better", 12% "moderately better", 5% "somewhat better", 5% "a little bit better", 6% "a tiny bit better" and 2% "about the same". Favorable changes in functional and psychological status were also noted, as shown by statistically significant improvements in measures of disability (Coefficient: 2.24; 95% credibility interval = 1.28 to 3.19; P = .99), function (Coefficient: 2.58; 95% credibility interval = 1.89 to 3.26; P = 1.00), fear of movement (Coefficient: -4.04; 95% credibility interval = -5.24 to -2.85; P = 1.00) and catastrophization (Coefficient: -7.41; 95% credibility interval = -9.00 to -5.82; P = 1.00) (**Table 2**).

## Association between changes in central pain processing and clinical and psychological outcomes

Improvements in the functioning of central pain processing mechanisms (CPM and TSP) following intervention were found to be very weakly associated with only few measures of clinical and psychological outcome as shown in **Table 3**. Improvement in the CPM response was

**Table 2. Effect of intervention on clinical, psychological and psychophysical variables.** Model adjusted for sex, age, BMI, individual heterogeneity (random effect) and baseline value of variable.

| Variables | Coefficient | 95% Credibility Interval | P |
|---|---|---|---|
| NDI (50) | -2.23 | 1.29 to 3.17 | 0.99 * |
| PCS (52) | -7.43 | -9.01 to -5.85 | 1.00 * |
| TSK (44) | -4.07 | -5.25 to -2.88 | 1.00 * |
| PSFS_mean | 2.58 | 1.89 to 3.26 | 1.00 * |
| Maximum pain 24h | -3.05 | -3.53 to -2.58 | 1.00 * |
| Mean pain 24h | -2.52 | -2.91 to -2.13 | 1.00 * |
| Pain on movement | -2.19 | -2.51 to -1.87 | 1.00 * |
| PPT ipsilateral trapezius | 0.22 | 0.03 to 0.42 | 0.98 * |
| PPT contralateral trapezius | 0.01 | -0.19 to 0.21 | 0.54 |
| PPT ipsilateral TA | -0.02 | -0.28 to 0.23 | 0.59 |
| PPT contralateral TA | 0.01 | -0.25 to 0.29 | 0.54 |
| TSP change (ratio) | -0.63 | -0.82 to -0.43 | 1.00 * |
| CPM change (absolute) | 0.89 | 0.14 to 1.65 | 0.99 * |

Abbreviations: BMI, body mass index; NDI, neck disability index; PCS, pain catastrophizing scale; TSK, Tampa scale kinesophobia; PSFS, Patient specific functional scale; PPT, pressure pain threshold; TA, tibialis anterior; TSP, temporal summation pain; CPM, conditioned pain modulation.

* Statistically significant change.

**Table 3. Association between changes in central pain processing mechanisms and clinical/psychological variables.** Model adjusted for sex, age, BMI and individual heterogeneity (random effect).

| | Change in CPM | | |
|---|---|---|---|
| | Coefficient | 95% Credibility Interval | P |
| **Change in NDI** | -0.57 | -1.75 to 0.59 | 0.83 |
| **Change in PCS** | -0.24 | -2.16 to 1.66 | 0.60 |
| **Change in TKS** | 0.43 | -1.01 to 1.88 | 0.72 |
| **Change in PSFS** | -0.65 | -1.22 to -0.07 | 0.98* |
| **Change in Max pain 24h** | 0.10 | -0.55 to 0.75 | 0.61 |
| **Change in Mean pain 24h** | -0.40 | -0.92 to 0.11 | 0.93 |
| **Change in Pain on movement** | -0.17 | -0.40 to 0.05 | 0.93 |
| **Change in GROC** | 0.28 | -0.29 to 0.87 | 0.83 |
| | Change in TSP | | |
| | Coefficient | 95% Credibility Interval | P |
| **Change in NDI** | 0.45 | -0.65 to 1.57 | 0.79 |
| **Change in PCS** | 0.18 | -1.74 to 2.10 | 0.57 |
| **Change in TKS** | 1.09 | -0.28 to 2.48 | 0.94 |
| **Change in PSFS** | -0.41 | -1.25 to 0.42 | 0.83 |
| **Change in Max pain 24h** | 0.22 | -0.38 to 0.83 | 0.76 |
| **Change in Mean pain 24h** | 0.21 | -0.25 to 0.69 | 0.82 |
| **Change in Pain on movement** | 0.42 | 0.10 to 0.74 | 0.99* |
| **Change in GROC** | -0.19 | -1.37 to 5.29 | 0.87 |

Abbreviations: BMI, body mass index; NDI, neck disability index; PCS, pain catastrophizing scale; TSK, Tampa scale kinesophobia; PSFS, Patient specific functional scale; PPT, pressure pain threshold; TA, tibialis anterior; TSP, temporal summation pain; CPM, conditioned pain modulation; GROC, global rating of change.

* Statistically significant at p < 0.05.

found to be negatively correlated with changes in PSFS (Coefficient: -0.65; 95% credibility interval = -1.22 to -0.07; P = 0.98), and attenuation of TSP was found to be associated with a greater improvement in pain during movement (Coefficient: 0.42; 95% credibility interval = 0.10 to 0.74; P = 0.99).

## Discussion

The present work provides novel evidence of restoration of normal central pain processing following manual therapy in NSCNP patients, as shown by TSP, CPM and PPT values returning to levels comparable to normative data collected in our lab [11]. In addition, clinical pain was ameliorated following treatment, and both functional and psychological measures were improved.

Significant attenuation of TSP was found here following manual therapy, an observation in keeping with previous reports using this therapeutic modality both in healthy volunteers [56] and pain conditions such as low back pain [57] and carpal tunnel syndrome [58]. It is noteworthy, however, that studies conducted so far specifically in patients with NSCNP have failed to report changes in TSP following a variety of interventions other than manual therapy, including virtual reality [22], cervical therapeutic exercise [22, 59] and a combined protocol of electrotherapy and cervical therapeutic exercise [15]. This raises an interesting question as to the ability of different therapy modalities to influence TSP, a scenario where manual therapy has proven to exert a significant attenuating effect as shown here and may therefore be best indicated in patients with a pronounced pronociceptive profile at baseline. Future studies using a wider range of therapeutic approaches are expected to provide further insights into this issue.

Alternatively, differences in stimulation techniques used to evoke the TSP response might have contributed to the disparity of results, considering that previous studies in NSCNP relied on mechanical stimuli as opposed to electrical stimuli as used here. Unfortunately, no comparative or validity studies are yet available in this regard, despite good reliability of various methodological approaches for assessing TSP [60–62]. Whereas mechanical stimuli used by previous studies such as weighted pinprick stimuli [21] or pressure exerted by either a cuff [59] or an algometer [22] stimulate superficial and deep mechanical nociceptors, electrical pulses largely bypass nociceptors to directly recruit afferent C-fibers [63]. Direct stimulation of C-fibers is considered as a robust mechanism to elicit the originally described wind-up phenomenon in spinal dorsal horn neurons in animal models, whose perceptual correlate is assessed by quantifying TSP [64, 65].

We found that manual therapy also restored a normal CPM response in our cohort. This finding is consistent with a previous study where neurodynamic treatment, i.e. a form of manual therapy, improved CPM in a similar patient population [16]. Indeed, several studies have shown a general normalizing effect of treatment on the CPM response both in NSCNP [20, 59, 66] and other pain conditions [67, 68] regardless of the therapeutic approach. It thus appears that restoration of the CPM response to normal levels is a frequent outcome of treatment, and little dependent on the type of treatment or target population.

The relationship between an increased TSP response and the level of perceived pain in the NSCNP population is not entirely clear. Although a systematic review with meta-analysis did find some association in a population of back pain patients [69], two recent case-control studies studying NSCNP failed to do so [11, 70]. Considering that the expression of wind-up at spinal cord neurons may be genetically encoded [71], an enhanced TSP may be viewed as an indication of higher propensity to developing pain hypersensitivity. In support of this notion, a number of studies have shown TSP to be a predictor of pain prospectively [72–74]. From a neurophysiological standpoint, temporal summation is considered as one of the initiating

neuroplastic mechanisms of central sensitization [75, 76]. On the other hand, wind-up as the correlate of TSP in animal models has also been found to be profoundly influenced by descending supraspinal modulation [77–79]. It thus seems reasonable to assume that attenuation of TSP following manual therapy as shown here may, at least in part, be mediated by recruitment of central pain modulatory mechanisms. This view is further supported by the enhancing effect of manual therapy on the CPM response in patients with NSCNP as shown here. The CPM paradigm is an experimental model to assess the functional state of the so-termed Diffuse Noxious Inhibitory Controls, a widespread modulatory mechanism arising from the brainstem and operating on second order, wide dynamic range neurons in the spinal dorsal horn via the spinal dorsolateral funiculi [80]. An enhanced CPM is considered to reflect greater efficacy of endogenous analgesia mechanisms and thus a more favorable position to control central excitation induced by incoming peripheral nociceptive input [81].

A comprehensive model of the underlying mechanisms of manual therapy based on current research in neuroscience has been postulated which emphasizes the importance of neurophysiological responses to the mechanical stimulus of a manual therapy intervention [82]. Our present study provides further support to that view in showing that both central pain processing and clinical status were indeed improved following manual therapy. Interestingly, however, we found no distinct associations between the two types of outcomes, suggesting that restoration of central pain processing to normal levels may contribute to general clinical improvement in parallel or rather independently from a number of other non-neural mechanisms. For instance, dynamic magnetic resonance imaging has revealed improvements in local cervical segmental mobility following treatment with posterior-to-anterior cervical mobilizations [83–85], enhanced muscle function [47, 86, 87] and attenuation of stiffness at specific spinal segments [88–90] has also been observed following treatment with passive spinal mobilizations. Such interventions have also been associated with changes in sympatoexcitatory activation [48, 86, 91, 92] and a reduction in the concentration of inflammatory markers such as substance P [93]. In addition, several studies have reported significant within-group changes following sham manual therapy treatments [46–48, 94] which also suggests the involvement of contextual mechanisms. None of the above studies evaluated concomitant changes in central pain processing. Collectively, the available evidence provides support to the view that the mechanism of action of manual therapy may be inherently multifactorial. This rather broad range multiplicity of mechanisms may, as suggested by our present study, include the restoration of both pro-nociceptive and anti-nociceptive central pain processing mechanisms to normal levels.

On the other hand, it is possible that the characteristics of our sample may have contributed, at least in part, to this observed lack of association. Thus, although central pain processing mechanisms were indeed significantly altered prior to treatment in the study sample as compared to controls [11], those baseline alterations and clinical status were both modest [95], rendering any possible associations between improvements in both types of outcomes after therapy rather hard to detect.

## Limitations

Since no control group was used, changes in both clinical and pain measures noted here cannot unequivocally be attributed to the intervention. Factors such as the natural course of the condition and the potential influence of the placebo effect need to be considered. Nonetheless, no study has thus far reported spontaneous normalization of central pain processing, and the fact that clinical improvement following the intervention was achieved after three months of persisting clinical manifestations renders an alternative explanation rather unlikely.

The generalizability of our findings is constrained by the specificity of our studied population within the NSCNP, which comprised individuals who sought treatment in a private clinical setting and exhibited only mild clinical symptoms. Therefore, caution should be exercised when extrapolating these results to broader or more severe NSCNP populations.

The present study only measured outcomes in the short term, and thus whether the observed changes are long lasting was not determined.

## Conclusion

The physiological mechanisms underlying the clinical effect of manual therapy on NSCNP remain unclear. Collectively, however, the present observed beneficial effects on TSP and CPM responses support the notion that manual therapy may operate, to some extent, by influencing central pain processing to ameliorate pain and improve clinical status. Nonetheless, the fact that no clear association was observed between restoration of normal central pain processing and clinical outcome suggests a plurality of underlying mechanisms that may also likely involve biomechanical, physiological and psychological changes. Further studies are needed to determine specifically which mechanisms of action influence the clinical improvement of NSCNP with manual therapy.

## Supporting information

**S1 Checklist. CONSORT 2010 checklist of information to include when reporting a randomised trial\*.**
(DOC)

**S2 Checklist. STROBE statement—Checklist of items that should be included in reports of *cohort studies*.** Checklist annotated according to the manuscript, "Restoration of normal central pain processing following manual therapy in nonspecific chronic neck pain".
(DOCX)

**S1 File.**
(PDF)

**S1 Data.**
(SAV)

## Acknowledgments

The authors would like to express their gratitude to Hiru Fisioterapia for generously providing their facilities for conducting the present study.

## Author Contributions

**Conceptualization:** Josu Zabala Mata, Jon Jatsu Azkue, Maialen Araolaza Arrieta, Ion Lascurain-Aguirrebeña.

**Data curation:** Josu Zabala Mata, Estíbaliz Dominguez López.

**Formal analysis:** Marc Saez.

**Investigation:** Josu Zabala Mata, Jon Jatsu Azkue, Estíbaliz Dominguez López.

**Methodology:** Josu Zabala Mata, Jon Jatsu Azkue, Ion Lascurain-Aguirrebeña.

**Project administration:** Josu Zabala Mata.

**Resources:** Jon Jatsu Azkue.

**Software:** Jon Jatsu Azkue.

**Supervision:** Jon Jatsu Azkue, Joel E. Bialosky, Maialen Araolaza Arrieta, Ion Lascurain-Aguirrebeña.

**Validation:** Joel E. Bialosky.

**Writing – original draft:** Josu Zabala Mata.

**Writing – review & editing:** Jon Jatsu Azkue, Joel E. Bialosky, Ion Lascurain-Aguirrebeña.

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
