## [Decision Letter · Decision Letter 0]

19 Dec 2023

PONE-D-23-34005Restoration of normal central pain processing following manual therapy in nonspecific chronic neck painPLOS ONE

Dear Dr. Zabala,

Thank you for submitting your manuscript to PLOS ONE. After careful consideration, we feel that it has merit but does not fully meet PLOS ONE’s publication criteria as it currently stands. Therefore, we invite you to submit a revised version of the manuscript that addresses the points raised during the review process.

**ACADEMIC EDITOR: **Three expert reviewers assessed the manuscript. Although they have considered this manuscript, they have raised some significant comments and feedback. In my opinion, the inclusion of those comments will greatly enhance the quality of this manuscript. Furthermore, I have a few comments and recommendations as well.

We look forward to receiving your revised manuscript.

Kind regards,

Shahnawaz Anwer, PhD

Academic Editor

PLOS ONE

Journal Requirements:

Reviewers' comments:

Reviewer's Responses to Questions

**Comments to the Author**

1. Is the manuscript technically sound, and do the data support the conclusions?

Reviewer #1: Yes

Reviewer #2: Partly

Reviewer #3: No

2. Has the statistical analysis been performed appropriately and rigorously? 

Reviewer #1: Yes

Reviewer #2: Yes

Reviewer #3: Yes

3. Have the authors made all data underlying the findings in their manuscript fully available?

Reviewer #1: Yes

Reviewer #2: Yes

Reviewer #3: No

4. Is the manuscript presented in an intelligible fashion and written in standard English?

Reviewer #1: Yes

Reviewer #2: Yes

Reviewer #3: Yes

5. Review Comments to the Author

Reviewer #1: A single cohort study was conducted which aimed to investigate whether a 4-week manual therapy treatment intervention restores normal functioning of central pain processing mechanisms in non-specific chronic neck pain. In addition, the relationship between changes in pain processing mechanisms and clinical outcome was assessed. The study showed an increase in CPM response and TSP. Weak associations were found between TSP/CPM measures and clinical outcomes.

Minor revisions:

1- Abstract: State the statistical testing methods and summary statistics or p-values to support the results.

2- Line 111: Indicate the statistical method which achieves 80% power. The power calculation should include: (1) the estimated outcomes in each group; (2) the α (type I) error level; (3) the statistical power (or the β (type II) error level); (4) the target sample size and (5) the statistical testing method and (6) for continuous outcomes, the standard deviation of the measurements.

3- The standard statistical term for average is mean.

4- Line 197: Label this section, “Statistical Analysis”.

5- Line 197: Indicate the underlying covariance structure used in the generalized linear mixed model and the criteria for selecting it.

6- Table 2: In addition to the frequency, provide the percentage female.

7- Table 2: Indicate if the underlying distribution of the data in Table 2 was checked for normality.

Reviewer #2: Reviewer's Report:

Title: Restoration of normal central pain processing following manual therapy in nonspecific chronic neck pain.

Abstract:

1. Lack of Specific Results: The abstract mentions that "an increased CPM response and attenuated TSP were found," but it does not provide specific quantitative results or effect sizes, making it difficult for readers to assess the significance of these findings.

2. Minor Associations: The abstract states that "only minor associations were found between normalization of TSP/CPM and measures of clinical outcome," which raises questions about the clinical significance of the observed changes in central pain processing.

3. Limited Information on Participants: It would be helpful to include some basic demographic information about the participants in the abstract to provide context for the study findings.

Introduction:

The introduction section provides valuable background information on the prevalence and challenges associated with non-specific chronic neck pain (NSCNP). However, there are some points to consider:

1. Lack of Clarity on Research Gap: While the introduction highlights the need for better understanding the mechanisms of manual therapy, it does not clearly specify the research gap or the specific questions the study aims to address.

2. Reference to Guidelines: The introduction mentions guidelines advocating exercise and manual therapy for NSCNP but does not provide specific references or citations, which could enhance the credibility of the claims.

3. Lengthy Background Information: The introduction contains extensive background information on the prevalence of neck pain, which, while informative, could be condensed for brevity.

Methods:

1. Sample Size Justification: The rationale for the sample size of 63 participants is based on the detection of a maximum difference of 10% in TSP and CPM measures, but it would be beneficial to provide more information on how this specific effect size was determined.

2. Lack of Control Group: The study lacks a control group, which makes it challenging to establish causation and attribute the observed changes solely to the manual therapy intervention.

3. Treatment Description: The methods describe the manual therapy intervention broadly but do not provide specific details about the techniques used or the treatment protocol, which limits the ability to replicate the study.

4. Statistical Analysis: While the statistical approach is mentioned, the methods could benefit from a more detailed explanation of the statistical models used and the rationale for choosing the Bayesian approach.

Results and Discussion

1. Lack of Control Group: One of the major limitations of this study is the absence of a control group. Without a control group, it is challenging to attribute the observed improvements in central pain processing and clinical outcomes solely to the manual therapy intervention. It is essential to account for the natural course of the condition and any potential placebo effects.

2. Weak Associations: The study reports very weak associations between changes in central pain processing mechanisms and clinical/psychological variables. This suggests that other factors or mechanisms might be at play in explaining the clinical improvement following manual therapy. The authors acknowledge this limitation but do not provide a more in-depth discussion on possible alternative explanations or mechanisms.

3. Methodological Differences: The authors mention methodological differences in stimulus types used in previous studies and the current study for evaluating central pain processing (TSP). However, the significance and potential implications of these differences are not discussed in detail in the Discussion section.

4. Implications for Clinical Practice: The manuscript could benefit from a more comprehensive discussion of the clinical implications of the findings. How can these results inform clinical practice, and what recommendations can be made for manual therapy in the treatment of NSCNP patients?

5. Sample Characteristics: The manuscript briefly mentions that the sample presented only mild baseline disability. This should be discussed more thoroughly, as it may have implications for the generalizability of the findings to the broader NSCNP population.

6. Limitations: The limitations section is somewhat brief. It would be beneficial to provide a more extensive discussion of the study's limitations, including the potential impact of the lack of a control group and the generalizability of the findings.

Conclusion: Authors did not written conclusion

Reviewer #3: The manuscript aims to investigate whether a 4-week manual therapy treatment restores normal central pain processing in non-specific chronic neck pain. While the topic is clinically relevant, the current manuscript requires clarity improvements to effectively convey necessary information to readers.

Language: Some sentences are lengthy and intricate, and simplifying them would enhance readability.

Title: The title could be more concise and directly convey the primary study outcome.

Abstract: It is recommended to specify the exact manual therapy techniques used in the abstract.

Introduction:

The references to previous studies in the manuscript lack specificity and detail regarding their methodologies and findings. The introduction and previous literature sections lack explicit mention of the manual therapy techniques employed in these referenced studies. Including specific details about the manual therapy techniques used in relevant studies is essential for providing context and understanding the existing body of literature. This addition would contribute to a more comprehensive and informed discussion of the background and rationale for the current study.

Method:

Inclusion/exclusion criteria should be presented in the text.

The rationale for choosing these specific techniques, their potential benefits for NSCNP, and details about why only one session was utilized need further clarification, ideally with the inclusion of figures.

The table format does not adhere to scientific writing standards, and

Include detailed information on questionnaire reliability and validity. If translated into your language, provide references for reliability and validity.

Discussion:

The discussion mainly repeats results without in-depth analysis or comparisons with previous literature. More extensive discussion on methodology, manual therapy techniques, and clinical applications is needed. The conclusion lacks clarity, and future research directions should be elaborated upon.

6. PLOS authors have the option to publish the peer review history of their article (what does this mean?). If published, this will include your full peer review and any attached files.

Reviewer #1: No

Reviewer #2: No

Reviewer #3: **Yes: **Sahar Boozari

---

## [Author Response · Author response to Decision Letter 0]

15 Feb 2024

1. Submission requirements

1.1. PLOS ONE's style requirements

- An asterisk has been added to the corresponding author (page 1, line 3).

- Authors' complete affiliations have been included (page 1, line 7-24).

- The current address of the corresponding author has been included (page 1, line 23-24).

- The name for the picture file has been changed (submission process).

1.2. Data Availability

All necessary data to replicate the results have been uploaded as Supporting Information (submission process).

1.3. Captions for your Supporting Information files

Captions have been included for the Supporting Information files at the end of our manuscript (page 31, lines 688-695), and in-text citations have been updated accordingly. 

2. Editors’ and reviewers’ comments

2.1. Title 

Reviewers' comment 

The title could be more concise and directly convey the primary study outcome.

Authors' response 

We respectfully disagree, as we believe the current title express the primary study outcome as directly as possible. We have carefully considered replacing "central pain processing" with terms such as "central pro- and antinociceptive mechanisms" or "temporal summation and conditioned pain modulation"; however, are concerned this would come across as wordy. Nevertheless, we will be happy to amend it if the editor or reviewers consider it appropriate. 

2.2. Abstract

2.2.1 Reviewers' comment

 State the statistical testing methods and summary statistics or p-values to support the results.

Authors' response 

Statistical testing methods (page 2-3, lines 43-46) and summary statistics (page 3, lines 47-51) were added.

2.2.2. Reviewers' comment

Lack of Specific Results: The abstract mentions that "an increased CPM response and attenuated TSP were found," but it does not provide specific quantitative results or effect sizes, making it difficult for readers to assess the significance of these findings.

Authors' response

Specific quantitative data have now been added (page 3, lines 47-51).

2.2.3. Reviewers' comment

Minor Associations: The abstract states that "only minor associations were found between normalization of TSP/CPM and measures of clinical outcome," which raises questions about the clinical significance of the observed changes in central pain processing.

Authors' response

We agree with the reviewer in that the strength of the association is weak; hence, rather than stating that the observed associations were of high clinical relevance, we stated that our data "suggests multiple mechanisms of action of manual therapy". We feel this statement is an accurate reflection of the results, in that it acknowledges some association between clinical outcome and normalization of pain processing (albeit small) but at the same time clearly states that mechanisms of manual therapy are most likely multifactorial in light of the small association we have found.

2.2.4. Reviewers' comment

Limited Information on Participants: It would be helpful to include some basic demographic information about the participants in the abstract to provide context for the study findings.

Authors' response

We have added gender and age characteristics of our sample (page 2, lines 37-38).

2.2.5. Reviewers' comment

It is recommended to specify the exact manual therapy techniques used in the abstract.

Authors' response

We have added a more detailed description of the manual therapy techniques used in the intervention (page 2, line 39).

2.3. Introduction 

2.3.1. Reviewers' comment

Lack of Clarity on Research Gap: While the introduction highlights the need for better understanding the mechanisms of manual therapy, it does not clearly specify the research gap or the specific questions the study aims to address.

Authors' response 

No study has so far assessed the effect of manual therapy on central pain processing mechanisms in NSCNP, and no prior study has explored associations between improvements in central pain processing and improvements in clinical status after manual therapy. Both are research gaps that our present study aims to address. We have modified the second part of the Introduction section in order to convey this issue more clearly. 

2.3.2. Reviewers' comment

Reference to Guidelines: The introduction mentions guidelines advocating exercise and manual therapy for NSCNP but does not provide specific references or citations, which could enhance the credibility of the claims.

Authors' response 

Additional references to guidelines have been included (4,5).

2.3.3. Reviewers' comment

Lengthy Background Information: The introduction contains extensive background information on the prevalence of neck pain, which, while informative, could be condensed for brevity.

Authors' response 

We have removed the following sentence on the prevalence of neck pain; “Moreover, recurrence reaches 50-75% within the next 5 years following the first episode, (3,4) and 68% of individuals experiencing an episode of acute neck pain will become chronic neck pain sufferers(5).”

2.3.4. Reviewers' comment

The references to previous studies in the manuscript lack specificity and detail regarding their methodologies and findings. The introduction and previous literature sections lack explicit mention of the manual therapy techniques employed in these referenced studies. Including specific details about the manual therapy techniques used in relevant studies is essential for providing context and understanding the existing body of literature. This addition would contribute to a more comprehensive and informed discussion of the background and rationale for the current study. 

Authors' response 

We have included additional information and details regarding manual therapy techniques in the treatment of NSCNP (page 4, lines 68-69) and other pathologies (page 5, lines 91-92) . 

2.4. Methods

2.4.1. Reviewers' comment 

Line 111: Indicate the statistical method which achieves 80% power. The power calculation should include: (1) the estimated outcomes in each group; (2) the α (type I) error level; (3) the statistical power (or the β (type II) error level); (4) the target sample size and (5) the statistical testing method and (6) for continuous outcomes, the standard deviation of the measurements.

Authors' response 

Further information has been added in the revised version of the manuscript.

2.4.2. Sample Size Justification: The rationale for the sample size of 63 participants is based on the detection of a maximum difference of 10% in TSP and CPM measures, but it would be beneficial to provide more information on how this specific effect size was determined.

Authors' response 

We have corrected an error (it should be “minimum” rather than “maximum”) and provided further information in this regard.

2.4.3. Reviewers' comment 

Lack of Control Group: The study lacks a control group, which makes it challenging to establish causation and attribute the observed changes solely to the manual therapy intervention.

Authors' response 

We acknowledge that this is a limitation of the study; as such we have commented on this issue in the discussion, in a subsection named "Limitations". Since there is no previous evidence of spontaneous normalization of central pain processing in such a small period of time (4 weeks) in chronic pain patients, we consider that the effects are most likely caused by the intervention (manual therapy). Nevertheless, in light of the reviewer´s comment, we have now further commented (limitations section) on the fact that changes could be due to the natural course of the condition and the potential influence of the placebo effects. 

2.4.4. Reviewers' comment 

The standard statistical term for average is mean.

Authors' response 

The term “average” has been replaced by "mean”.

2.4.5. Reviewers' comment 

Line 197: Label this section, “Statistical Analysis”.

Authors' response 

This section has been labeled as suggested. 

2.4.6. Reviewers' comment 

Line 197: Indicate the underlying covariance structure used in the generalized linear mixed model and the criteria for selecting it.

Authors' response 

Further information has been added in the revised version of the manuscript.

2.4.7. Reviewers' comment 

Treatment Description: The methods describe the manual therapy intervention broadly but do not provide specific details about the techniques used or the treatment protocol, which limits the ability to replicate the study.

Authors' response 

We have provided greater details about the treatment techniques in the intervention section (pages 10-11).

2.4.8. Reviewers' comment 

Statistical Analysis: While the statistical approach is mentioned, the methods could benefit from a more detailed explanation of the statistical models used and the rationale for choosing the Bayesian approach.

Authors' response 

We appreciate the reviewer’s comment and the opportunity to more strongly justify our analytic approach. As stated in the original submission of the manuscript (page 10, lines 201 to 204) models present great complexity because of data variability. First, there is individual heterogeneity, which comprises unobserved invariant variables over time that are specific to each individual participant, i.e. residual confounding. Second, each subject has at least two observations (in most cases three), and such non-independence must also be controlled for. 

Each observation must allow estimating three sources of variability, the parameters (i.e., the model coefficients) and the variances of the random effects (individual heterogeneity and dependence). This cannot be done using non-Bayesian methods. We have provided further information in the revised version of the manuscript (page 11).

2.4.9. Reviewers' comment 

Inclusion/exclusion criteria should be presented in the text.

Authors' response 

The inclusion/exclusion criteria have been added to the text (page 7, line 130-136), and the corresponding table has been removed in order to avoid replication of information. 

2.4.10. Reviewers' comment 

The rationale for choosing these specific techniques, their potential benefits for NSCNP, and details about why only one session was utilized need further clarification, ideally with the inclusion of figures.

Authors' response 

Patients received four 45-minute sessions of manual therapy, once a week for 4 weeks. Careful consideration was given to the type and dosage of manual therapy included in the current study. Our dosage aligns with many other studies of manual therapy (1–5) and the included interventions have previously been found effective for the treatment of patients with NSCNP (6–11). We have incorporated in the manuscript (intervention section, page 10) additional references justifying our selected approach. 

2.4.11. Reviewers' comment 

The table format does not adhere to scientific writing standards.

Authors' response 

We thank the reviewer for bringing this to our attention. We have revised the format of all tables to ensure consistency and alignment with established scientific writing conventions. 

2.4.12. Reviewers' comment 

Include detailed information on questionnaire reliability and validity. If translated into your language, provide references for reliability and validity.

Authors' response 

We have provided further information regarding the validity and reliability of questionnaires (pages 7-8). 

2.5. Results and Discussion 

2.5.1. Reviewers' comment

Table 2: In addition to the frequency, provide the percentage female.

Authors' response 

The percentage female has been provided (Table 2 of the original submission is now Table 1). 

2.5.3. Reviewers' comment

Table 2: Indicate if the underlying distribution of the data in Table 2 was checked for normality.

Authors' response 

Although the dependent variables are continuous, the small sample size leads to the rejection of the null hypothesis of normality in all cases (Shapiro-Wilk test). However, and as a sensitivity analysis, we repeated the main analyzes using, as an alternative link function, the normal-inverse Gaussian(12). These types of models are a flexible extensions of Gaussian models because they contain the Gaussian model as a special case. Using these models, the results were practically the same.

2.5.4. Reviewers' comment

Lack of Control Group: One of the major limitations of this study is the absence of a control group. Without a control group, it is challenging to attribute the observed improvements in central pain processing and clinical outcomes solely to the manual therapy intervention. It is essential to account for the natural course of the condition and any potential placebo effects. 

Authors' response 

As discussed above in point 2.4.3, we acknowledge that this is a limitation of the study and comment on this in the section named “limitations”. Furthermore, in this section we have now further commented on the fact that changes could be due to the natural course of the condition and the potential influence of the placebo effects.

2.5.5. Reviewers' comment

Weak Associations: The study reports very weak associations between changes in central pain processing mechanisms and clinical/psychological variables. This suggests that other factors or mechanisms might be at play in explaining the clinical improvement following manual therapy. The authors acknowledge this limitation but do not provide a more in-depth discussion on possible alternative explanations or mechanisms.

Authors' response 

In the last paragraph of the Discussion section of the present revised version we now provide two avenues to interpret this finding. Firstly, a straightforward interpretation is seeing our results as suggesting other alternative mechanisms at play. Since confirmatory evidence regarding possible mechanisms is rather limited, we can only speculate by enumerating potential mechanisms supported by little literature, which we have in the present version. Secondly, the occurrence of weak associations might be related to characteristics of the sample, which showed only mild alterations at baseline. The latter may be seen as a limitation to the study and would justify further studies in neck pain patients with a broader clinical presentation. 

2.5.6. Reviewers' comment

Methodological Differences: The authors mention methodological differences in stimulus types used in previous studies and the current study for evaluating central pain processing (TSP). However, the significance and potential implications of these differences are not discussed in detail in the Discussion section. 

Authors' response 

We have added a more detailed discussion about the methodological differences (page 19, lines 329-338). 

2.5.7. Reviewers' comment

Implications for Clinical Practice: The manuscript could benefit from a more comprehensive discussion of the clinical implications of the findings. How can these results inform clinical practice, and what recommendations can be made for manual therapy in the treatment of NSCNP patients?

Authors' response 

We have delved more deeply into this topic at the end of the discussion section, however, since our study is grounded in fundamental research focused on understanding mechanisms of action of manual therapy, the possibility of making direct immediate recommendations for clinical practice are limited.

2.5.8. Reviewers' comment

Sample Characteristics: The manuscript briefly mentions that the sample presented only mild baseline disability. This should be discussed more thoroughly, as it may have implications for the generalizability of the findings to the broader NSCNP population.

Authors' response 

This has been discussed in the limitations section.

2.5.9. Reviewers' comment

Limitations: The limitations section is somewhat brief. It would be beneficial to provide a more extensive discussion of the study's limitations, including the potential impact of the lack of a control group and the generalizability of the findings.

Authors' response 

In addition to expanding on the limitations related to the lack of a control group, we have included a paragraph addressing the constraints on the generalizability of our results (page 21, lines 396-399).

2.5.10. Revi

---

## [Decision Letter · Decision Letter 1]

19 Mar 2024

PONE-D-23-34005R1Restoration of normal central pain processing following manual therapy in nonspecific chronic neck painPLOS ONE

Dear Dr. Zabala,

Thank you for submitting your manuscript to PLOS ONE. After careful consideration, we feel that it has merit but does not fully meet PLOS ONE’s publication criteria as it currently stands. Therefore, we invite you to submit a revised version of the manuscript that addresses the points raised during the review process.

**ACADEMIC EDITOR:**Dear Authors!

While your revised manuscript read better, there are still some important comments raised by the reviewers. Please address all the reviewer comments carefully.

We look forward to receiving your revised manuscript.

Kind regards,

Shahnawaz Anwer, PhD

Academic Editor

PLOS ONE

Journal Requirements:

Additional Editor Comments:

Dear Authors!

While your revised manuscript read better, there are still some important comments raised by the reviewers. Please address all the reviewer comments carefully.

Reviewers' comments:

Reviewer's Responses to Questions

**Comments to the Author**

1. If the authors have adequately addressed your comments raised in a previous round of review and you feel that this manuscript is now acceptable for publication, you may indicate that here to bypass the “Comments to the Author” section, enter your conflict of interest statement in the “Confidential to Editor” section, and submit your "Accept" recommendation.

Reviewer #1: (No Response)

Reviewer #2: (No Response)

Reviewer #3: All comments have been addressed

2. Is the manuscript technically sound, and do the data support the conclusions?

Reviewer #1: Yes

Reviewer #2: Yes

Reviewer #3: Yes

3. Has the statistical analysis been performed appropriately and rigorously? 

Reviewer #1: Yes

Reviewer #2: Yes

Reviewer #3: Yes

4. Have the authors made all data underlying the findings in their manuscript fully available?

Reviewer #1: Yes

Reviewer #2: Yes

Reviewer #3: No

5. Is the manuscript presented in an intelligible fashion and written in standard English?

Reviewer #1: Yes

Reviewer #2: Yes

Reviewer #3: Yes

6. Review Comments to the Author

Reviewer #1: Only one prior comment was not adequately addressed. in the sample size justification section, indicate the statistical testing METHOD which achieves 80% power.

Reviewer #2: Restoration of normal central pain processing following manual therapy in nonspecific chronic neck pain. Manuscript is written very well, methodology need more detail, study findings are very interesting There are a few points that need to be addressed. The study is good and very helpful for clinical Physical Therapy Health Professionals.

1- Abstract: Revised abstract sound good.

2- What was reason of high percentage of female participants with NSCNP, is there any reference supporting high prevalence of pain in female.

3- Methodology: Need more detailed procedure for manual therapy, like who did this procedure of mobilization, is it the same researcher, who evaluated pre-and post intervention outcomes or other, was he certified manual therapist with how many years of experience?

4- P-A mobilization with how many, thrust or oscillations in each direction.

5- Mobilization force was applied at spinous process or at transvers process, if so how did you identified or located these landmarks.

6- While giving mobilization force vertebra was stabilized for example C4-C5 vertebra there is hypomobility.

7- How did authors gave A-P mobilizations at cervical spine.

8- Please attach few pictures of manual therapy, placement of hands, direction of mobilization etc..

9- How did authors calculated sample size (63), please write formula.

10- How reliable and valid was the Questionnaire used in this study?

11- Findings are very interesting.

Thanks & Regards.

Reviewer #3: (No Response)

7. PLOS authors have the option to publish the peer review history of their article (what does this mean?). If published, this will include your full peer review and any attached files.

Reviewer #1: No

Reviewer #2: No

Reviewer #3: **Yes: **Sahar Boozari

---

## [Author Response · Author response to Decision Letter 1]

9 Apr 2024

We are once again grateful for your thorough review. We are committed to implementing all of your suggestions to improve the quality of the article. Below, we provide explanations for the points raised.

 Editors’ and reviewers’ comments

 Reviewers' comment 

In the sample size justification section, indicate the statistical testing METHOD which achieves 80% power.

Authors' response 

We used the 'pwr' package, in the free statistical environment R (version 4.3.2)(1,2), based on the formula provided by Cohen:

Power=h√(n/2)-z_(1-α⁄2)

Where z_(1-α⁄2) is the normal curve α/2th percentile; h=ϕ_1-ϕ_2; and ϕ=2 arcsin⁡√p; where p denoted the proportion (i.e., p1 of the treatment group, and p2 of the treatment group).

 Reviewers' comment 

What was reason of high percentage of female participants with NSCNP, is there any reference supporting high prevalence of pain in female. 

Authors' response 

The literature consistently demonstrates a higher prevalence of neck pain among women(3–5), even emerging as a risk factor in its own right(6). Our recruitment process involved enrolling patients as they presented with neck pain at our clinic, without making any exceptions based on gender. This approach ensured that our study population closely mirrors the gender distribution observed for this condition. 

 Reviewers' comment 

Methodology: Need more detailed procedure for manual therapy, like who did this procedure of mobilization, is it the same researcher, who evaluated pre-and post intervention outcomes or other, was he certified manual therapist with how many years of experience?

Authors' response 

All treatments were administered by the same therapist, who is also the primary researcher. As indicated in the intervention section, the therapist possesses postgraduate training and over 15 years of experience in musculoskeletal physiotherapy. Evaluation of both clinical and neurophysiological outcomes was conducted by another researcher, Estibaliz Dominguez. This setup ensured that the therapist administering the treatments remained blinded to the baseline and post-treatment measures. This supplementary information has been included in the intervention section (page 11, lines 222-224).

 Reviewers' comment 

P-A mobilization with how many, thrust or oscillations in each direction.

Authors' response 

To provide clarification regarding manual therapy passive mobilizations, we have included the following paragraph in the intervention section: “All passive mobilizations were performed using oscillatory techniques, comprising sets of 6 oscillations. The procedure continued until the hypomobile segments regained motion, or alternatively, a maximum of 4 sets was reached. The direction and intensity of the technique were determined by the clinician based on prior clinical assessment.”

 Reviewers' comment 

Mobilization force was applied at spinous process or at transvers process, if so how did you identified or located these landmarks. 

Authors' response 

We conducted mobilizations both at spinous process and transvers process, depending on the intended purpose of the technique, whether it was unilateral or bilateral. The identification of these landmarks was achieved based on the therapist anatomical and clinical expertise.

 Reviewers' comment 

While giving mobilization force vertebra was stabilized for example C4-C5 vertebra there is hypomobility.

Authors' response 

This depended on the technique employed; in some instances, the patient's degree of flexion was utilized to stabilize one portion of the spine, while in others, the therapist's non-mobilizing hand provided stabilization. Specifically, in the case of bilateral hypomobility between C4-C5, a posterior-to-anterior mobilization in the prone position did not require specific stabilization; instead, force was applied directly to the spinous process of C4.

 Reviewers' comment 

How did authors gave A-P mobilizations at cervical spine.

Authors' response 

This technique was performed with the patient lying supine, with the therapist accessing the transverse processes from the front of the neck and gently applying antero-posterior forces.

 Reviewers' comment 

Please attach few pictures of manual therapy, placement of hands, direction of mobilization etc.. 

Authors' response 

These techniques represent a subset of those employed in the study.

 Suboccipital Muscle Treatment 

Upper Trapezius Myofascial Trigger Point treatment

Upslope and Downslope Mobilisation

 ECOM Myofascial Trigger Point treatment 

Central Posterior-Anterior Mobilisation

Unilateral Anterior-Posterior Mobilisation 

 Reviewers' comment 

How did authors calculated sample size (63), please write formula. 

Authors' response 

The response to this comment has been previously addressed in point 1.1.

 Reviewers' comment 

How reliable and valid was the Questionnaire used in this study?

Authors' response 

In this study, three questionnaires were utilized: the Neck Disability Index (NDI), the Pain Catastrophizing Scale (PCS), and the Tampa Scale of Kinesiophobia (TSK). The reliability and validity of these questionnaires are detailed in the clinical assessment section (page 7, lines 138-151).

---

## [Editor Report · Decision Letter 2]

26 Apr 2024

Restoration of normal central pain processing following manual therapy in nonspecific chronic neck pain

PONE-D-23-34005R2

Dear Dr. Zabala,

We’re pleased to inform you that your manuscript has been judged scientifically suitable for publication and will be formally accepted for publication once it meets all outstanding technical requirements.

Kind regards,

Shahnawaz Anwer, PhD

Academic Editor

PLOS ONE

Additional Editor Comments (optional):

Authors are congratulated for their diligent work and significant revisions made based on the reviewer comments. Manuscript is significantly improved.
---

## [Editor Report · Acceptance letter]

3 May 2024

PONE-D-23-34005R2 

PLOS ONE

Dear Dr. Mata, 

I'm pleased to inform you that your manuscript has been deemed suitable for publication in PLOS ONE. Congratulations! Your manuscript is now being handed over to our production team.

Kind regards, 

on behalf of

Dr. Shahnawaz Anwer 

Academic Editor

PLOS ONE